# Charophytes (Charophyceae, Charales) of South Kazakhstan: Diversity, Distribution, and Tentative Red List

**DOI:** 10.3390/plants12020368

**Published:** 2023-01-12

**Authors:** Satbay Nurashov, Gaukhar Jumakhanova, Sophia Barinova, Roman Romanov, Elmira Sametova, Aibek Jiyenbekov, Saule Shalgimbayeva, Thomas Edward Smith

**Affiliations:** 1RSE on REM “Institute of Botany and Phytointroduction” FWLC MEGNR RK, 36 “D” Timiryazeva Str., Almaty 050040, Kazakhstan; 2Faculty of Biology and Biotechnology, Al-Farabi Kazakh National University, 71 Al-Farabi Ave., Almaty 050040, Kazakhstan; 3Institute of Evolution, University of Haifa, Abba Khoushi Ave, 199, Mount Carmel, Haifa 3498838, Israel; 4Komarov Botanical Institute of the Russian Academy of Sciences, Professora Popova Str. 2, 197376 St. Petersburg, Russia; 5Institute for Water and Environmental Problems, Siberian Branch of the Russian Academy of Sciences, Molodezhnaya Str. 1, 656038 Barnaul, Russia; 6Faculty Math and Science, Arkansas State University Beebe, 1000 W Iowa St., Beebe, AR 72012, USA

**Keywords:** Characeae, Kazakhstan, morphology, ecology, DNA barcoding, species protection

## Abstract

The presented research was conducted during 2019–2022 in south and southeast Kazakhstan to document the species richness, distribution, and ecology of charophytes (Characeae) as a first step towards to estimate the need for species protection. Across the 54 sites, we found ten species and one variety. *Chara vulgaris* Linnaeus and *C. contraria* A.Braun ex Kützing were the most common species, followed by *C. canescens* Loiseleur, *C. kirghisorum* C. F. Lessing, *C. tomentosa* Linnaeus, *C. dominii* J. Vilhelm, *C. globata* W. Migula, *Nitellopsis obtusa* (Desvaux) J. Groves, and *Nitella hyalina* (De Candolle) C. Agardh. The list of localities for each species was compiled. The distribution of each taxon was mapped in relations to the ecoregions studied. The two most frequent species were found in a wide spectrum of ecoregions, whereas all other species occurred in only a few regions in Kazakhstan. The Kaskelen River Valley had the most sampled sites with the highest number of co-occurring species (up to five together). Statistical maps were plotted in attempt to outline key environmental variables explaining the distribution of each species. A comparison of species and environmental variables distribution maps lets us assume that *C. vulgaris* prefers low altitude habitats with higher water temperatures, organic enrichments, and color, but low oxygen and pH. Other species prefer clear, alkaline, organically unpolluted, and well-oxygenated waters in lowland habitats. The redundancy detrended analysis (RDA) defined pH and altitude as negative factors for *Nitellopsis obtusa* whereas an increase in water temperature was positive. Altitude and water temperatures affected *Chara contraria* positively while altitude negatively influenced the rare species: *Chara tomentosa*, *C. kirghisorum*, and *C. dominii*. The *mat*K sequences were obtained for *C. contraria* and *C. vulgaris* to confirm their identity according to morphological traits and to compare populations of *C. gymnophylla* and *C. vulgaris* from an arid region in Israel. Our data allowed for the preparation of a tentative red list from the study region. One species was recognized as endangered, four species and one variety as vulnerable, and two species as least concern. There was insufficient data to determine the status of two species and one variety.

## 1. Introduction

Charophytes are members of the large phylum Charophyta [1] with six classes. Our study investigates the diversity from the family Characeae in the order Charales. This group is monophyletic [2] and consists of highly developed benthic macroalgae. Charales has received considerable taxonomic attention because they occupy an important place in the Tree of Life since the terrestrial plants originated from this group [3]. Charophytes are distributed throughout the world, except for Antarctica. They grow in freshwater lakes, streams, rivers, and wetlands. Some species are also found in brackish or saline waters [4].

Charophyte communities usually form monospecies mats or are found together with other macroalgae [5] and magnoliophytes. Frequently, charophytes are recognized as a pioneer species that colonize emerging or disturbed water bodies [6], but some of them are perennial winter green plants able to grow for an indefinite period in a stable environment. *Chara* and *Nitella* are widely distributed and relatively species-rich genera in comparison to the other genera of charophytes [7].

Species identification of *Chara vulgaris* and *C. contraria* by macro-features has long been a subject of debate among taxonomists. They have limited morphological characteristics, intermediate species, and an unclear degree of habitat or plant developmental pressure leading to morphological variability [7]. Both species are considered by most authors as separate *Chara* species, but at the same time, a high morphological similarity of both species is noted by R.D. Wood. He characterized *C. contraria* as a conspecific variety of *C. vulgaris* due to their similar morphologies [7]. They differ mainly in the structure of the cortex and the position of the spiny cells, while the cortex of *C. vulgaris* is defined as aulacanthous (secondary rows are more visible, spiny cells seem to be located in furrows), and *C. contraria* as tylacanthous (primary rows stand out, spine cells seem to sit on ridges). Both species are characterized as haplobiont, monoecious, reproducing predominantly by self-fertilization. Consequently, these species show extremely limited genetic variability compared to diploid forms. Experimental crosses between *C. vulgaris* and *C. contraria* have clearly shown that they are reproductively isolated [7]. Based on this evidence, molecular studies of the genome of both species have a critical role in taxonomic identification and help to determine the species-specific habitat parameters.

Charophyte diversity studies in Kazakhstan are centered in bodies of water in northern and southern territories, e.g., the deltas of the Ili, Syrdarya, and Amudarya rivers, southwest Siberian Plain, Saryarqa (Kazakh Upland), the Turgai depression, the lakes of Burabay National Park, and Lake Balkhash drainage basin [8]. There have only been a few studies and limited data from south, east, and west Kazakhstan. There have been only 28 species and three forms of charophytes, including 22 species and three forms of *Chara*, three species of *Nitella*, and one species of *Lamprothamnium*, *Lychnothamnus*, and *Nitellopsis* each found in previous studies in southern Kazakhstan [8,9,10,11,12,13,14,15,16,17,18,19,20,21]. The highest species richness (26 species and two forms) is concentrated in the lowlands of the Ili River drainage basin, especially the Ili River delta and water bodies neighboring Lake Balkhash. These recorded datasets are essential for the detection of the trends in occurrence and abundance of charophytes, but some of them are not precise enough for this estimation. There were 40 species and two forms of charophytes found in previous studies in Kazakhstan [8,22,23,24,25,26,27,28].

Recent studies have shown that water quality variables, such as temperature, pH, salinity, electrical conductivity, total dissolved solids, and nutrient saturation, affect charophyte spatial distribution, diversity, and ecology [29,30,31,32,33,34,35,36,37,38,39,40,41,42]. Nevertheless, the research on the effects of water quality on spatial distribution is yet to be undertaken in south and southeast Kazakhstan because these environmental relations might not be the same in different ecoregions, especially in the large arid regions from this study.

Human impact and management of water bodies resulting in significant environmental changes has led to a gradual decline in abundance, occurrence, and diversity of charophytes over the past decades throughout the regions [43,44,45,46,47,48]. As a result, some regional species have been recognized as endangered or even extinct and added to the Red List. Charophytes are one of the most sensitive and threatened plant groups [49,50,51,52]. The most severe threats to their survival would be expected in arid and semi-arid regions as a result of a combination of anthropogenic transformation in the environment and adverse climatic changes resulting in destruction and habitat loss. There has been a drastic decline in some species occurring in Lake Balkhash drainage basin below the Kapchagay Reservoir. This was observed during 1975–1978 because the Ili River discharge regulations changed, impacting the reservoir [53].

There are no species included in the national Red Data Book from Kazakhstan (https://www.inaturalist.org/projects/red-book-of-kazakhstan-plants, accessed on 20 May 2022). Therefore, we investigated the main threatened species during our study to identify charophyte diversity and species regional distribution. Since the two species of Characeae, *C. vulgaris* and *C. contraria*, are widespread and extremely similar in morphology, it was necessary to compare DNA sequences to determine speciation between the two species and subsequent comparison in the NCBI data.

In this investigation, we aimed to describe species of charophytes from the south and southeast Kazakhstan. We use ecomorphological and polyphasic approaches (i.e., genotypic, chemotaxonomic, and phenotypic methods) to determine the taxonomic position of organisms [54,55] and present the results on their diversity, ecology, relationship with the main environmental factors, and distribution as a first step towards their protection.

## 2. Results

### 2.1. Charophytes Diversity and Distribution

Altogether, there were 10 species and one variety of charophytes identified from 54 investigated regional sites (Appendix B). Taxon distributions were mapped in accordance to the ecoregion they were found (Figure 1a,b). From Figure A1, charophyte thickets usually formed far from the shoreline, apparently as a result of a significant fluctuation of the water levels in arid climates. Some sites were polluted which were confirmed by high BOD (Appendix A
Table A1) which inhibits the formation of large mats of charophytes.

*Chara vulgaris* (26 sites) and *C. contraria* (20 sites) were widely distributed in the regions (Figure 1e,g). These two species were found in two regions: b, Central Asian riparian woodlands, and f, Tian Shan montane steppe and meadows. At the same time, *C. canescens*, *C. kirghisorum*, *C. tomentosa*, *C. dominii*, *C. globata*, and *Nitellopsis obtusa* were concentrated in the central Asian riparian woodland ecoregion (Figure 1d,f,h). *Nitella hyalina* was found in two closely related ecoregions in southeast Kazakhstan: central Asian riparian woodlands and Emin Valley Steppe (Figure 1d). Only one species *Chara aspera* and its variety *subinermis* is widely distributed across the studied territory (Figure 1c). Moreover, this variety was found separately from the variety only in the central Asian riparian woodland ecoregion. The site with the largest species richness was Kaskelen River pond 1, in which there were five species: *Chara aspera, C. contraria, C. kirghisorum, C. vulgaris*, and *C. tomentosa*. The Kaskelen River is not large and located on the flat landscape where it is dammed and divided into three water bodies in the river delta close to the Kapchagai Reservoir. These ponded areas of the river delta were rich in species, such as *C. aspera*, *C. contraria*, *C. kirghisorum*, *C. tomentosa*, *C. dominii*, *C. globata*, and *C. vulgaris,* with addition of *C. vulgaris* and *C. contraria* in the upper reaches of the Kaskelen River. Therefore, the Ili River basin was the richest area and had the largest diversity of charophytes from the studied regions.

Specimens collected, housed in the Institute of Botany and Phytointroduction in Almaty, are presented in Appendix B, along with morphological descriptions and distribution data regarding the studied areas and herbarium numbers.

Statistical maps were constructed for each identified species with known environmental variables (Appendix A
Table A1 and Appendix A
Table A2). Figure 2a demonstrates mapping method suitability for the distribution of environmental variables, i.e., Ketmen Ridge’s mountain elevations can be recognized in the southern Almaty Region. From the maps, water temperature was highest in the south and central parts of Ili River basin situated in the desert zone (Figure 2b), BOD and water Pt/Co color also increased to the south (Figure 2c,d), while oxygen saturation and pH decreased (Figure 2e,f).

**Figure 1 plants-12-00368-f001:**
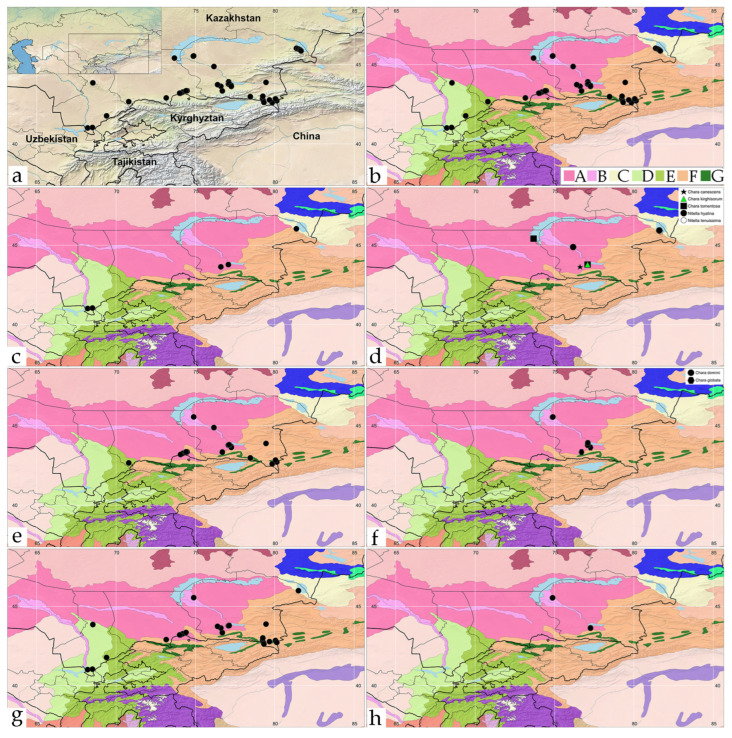
Species distribution in context of elevation (**a**) and ecoregions (**b**–**h**) in the regions studied: (**a**,**b**)—all species, (**c**)—*Chara aspera*, (**d**)—*C. canescens* (1), *C. kirghisorum* (2), *C. tomentosa* (3), *Nitella hyalina* (4), (**e**)—*C. contraria*, (**f**)—*C. dominii* (1), *C. globata* (2), (**g**)—*C. vulgaris*, (**h**)—*Nitellopsis obtusa*. Ecoregions [56]: A—Central Asian northern desert, B—Central Asian riparian woodlands, C—Emin Valley steppe, D—Alai-Western Tian Shan steppe, E—Gissaro-Alai open woodlands, F—Tian Shan montane steppe and meadows, G—Tian Shan montane conifer forests.

Species distribution maps were used to visualize and demonstrate the tendencies of each species in the study. *Chara aspera* and *C. tomentosa* preferred north and east regional environments (Figure 3a,b), whereas *C. dominii* and *C. globata* were found across the Ili River basin (Figure 3c,d). The tendency of two similar species was very interesting. *C. vulgaris* and *C. contraria* tended to differ in dissimilar environmental conditions and geographic areas. *C. vulgaris* occupied southern water habitats (Figure 3e) while *C. contraria* preferred more northern habitats (Figure 3f). Their distributions seemed not to be linked, as if mutually exclusive, even though both species sometimes occupy some habitats together.

Statistical maps of other *Chara* species demonstrates a limited distribution for *C. canescens* and *C. kirghisorum* (Figure 4a,b). *Nitella hyalina* and *Nitellopsis obtusa* (Figure 4c,d) were found in a few occupied areas. *Nitella hyalina* has a unique distribution, whereas *Nitellopsis obtusa’s* distribution was similar to that of *C. dominii*.

Comparing the species and environmental variables distribution maps allowed us to presume that *Chara vulgaris*, the most distributed species in south and southeast Kazakhstan, preferred low altitude habitats with high water temperatures, organic enrichments (increased values of BOD), and color, but low oxygen and pH. Other species, e.g., *C. aspera*, *C. tomentosa*, *C. dominii*, *Nitellopsis obtusa*, and *Nitella hyaline*, preferred clear, alkaline, organically unpolluted, and well-oxygenated waters in lowland habitats.

To clarify the factors influencing the distribution of species in Kazakhstan, RDA analysis was carried out, as detailed in Appendix A
Table A2. Average species richness was 11 species in 34 sites that are dependent on biological variables, whereas environmental variables of water and air temperature, pH, index of organic pollution S, and site altitude were independent variables.

There are species that are highly influenced by pH, temperature, and altitude, while organic pollution (Index S) is not an important factor for regulating species distributions (Figure 5). RDA shows that pH and altitude are negatively correlated to *Nitellopsis obtusa*’s distribution, but higher water temperatures are positively correlated. Increasing site altitude and water temperatures were positive influences for *Chara contraria*. At the same time, the habitat altitude is a negative factor for the rare species *Chara tomentosa*, *C. kirghisorum*, and *C. dominii’s* distributions. For the other species, the RDA does not indicate specific influencing factors.

Therefore, RDA helped to identify major environmental variables affecting each species, groups of species, and rare species’ preferences. The RDA results confirm the visualized distribution of species and environmental variable map comparisons, as well as provide some characteristics concerning the preferred environments of the rare species.

From the available data on the ecology of 38 species [57], it is known that they prefer waters slightly saturated with organic matter. They are more represented by indicators of oligosaprobic waters, where the saprobity index ranges from 0.8 to 1.3, and the trophic status has a wide amplitude from oligo- to eutrophic. Among the 11 species identified in Kazakhstan, only seven species are known to prefer organic pollution and trophic status. All preferred to exist in oligosaprobic communities with an index amplitude of 1.1–1.2 and a trophic status from oligo- to eutrophic waters. In our case, for the populations of the studied charophytes, the saprobity index varied in amplitude of 1.26–2.11, which shows adaptability to the more organically saturated waters in our study but from the same wide range of trophic status.

### 2.2. DNA Sequencing and Species Definition

DNA extractions taken from samples from the studied populations in Kazakhstan were compared to populations from Israel that had similar climates. DNA was isolated from two species (*Chara vulgaris* and *C*. *contraria*) from Kazakhstan and Israel. Thus, sequencing was carried out for the two most common species from climatically similar regions. In Table 1, environmental data for the DNA sequenced populations are presented for *Chara* species from two sites in south and southeast Kazakhstan and ten sites in Israel. For comparison with charophytes from Israel, we selected the *C. vulgaris* from seven habitats and the closely morphologically related *C. gymnophylla* A. Braun from three habitats, but *C. contraria* from this region has not yet been sequenced.

Figure 6 demonstrates high similarity of *mat*K sequences of *Chara vulgaris* samples from the Kakpatas River (yellow) with other populations of *C*. *vulgaris* from NCBI. It confirms the identity of *C. vulgaris* according to morphological traits only from different habitats, which includes NCBI data in the tree that is very closely related to our data.

Figure 7 reflects a high similarity of *mat*K sequences for *Chara contraria* samples from the Anniversary Lake (yellow) and populations from other regions according to NCBI data. These data confirm our identification of *C. contraria* based on morphological traits and sequencing data.

The phylogenetic tree presents a high similarity of *mat*K sequences for *Chara contraria, C. vulgaris,* and *C. gymnophylla* samples from the south and southeast Kazakhstan and Israel sites in Figure 8. There are a few clusters with high similarity of sequences shown in Table 1. Most similar were populations of *C. vulgaris* from Mediterranean coastal zone with high salinity and pH (Table 1). *C. vulgaris* populations from Oren can be included in cluster 1 too. *C. gymnophylla* was a slightly different form from northern Israel (Dafna) that is in cluster 2. Cluster 3 combined *C. vulgaris* populations from Carmel Mountain Biosphere Reserve and Ein Afeq Natural Reserve, Israel and *C. gymnophylla* from the Upper Jordan River valley habitat Ein Tao. High mountain habitat Nevoria in northern Israel included *C. gymnophylla* in cluster 4. *C. vulgaris* from Kakpatas River in Kazakhstan and Ein El Verde from the mountainous region of northern Israel was grouped into cluster 5. The last, significantly different cluster 6 included only one population of *C. contraria* from the Lake Anniversary in Kazakhstan.

We analyzed the environmental variables and their importance to regulate the species into clusters but only found similar parameters for cluster 1. *C. vulgaris* species was found from coastal zone sites of Israel where salinity was highest (Table 1). All other sites differed in salinity, pH, nitrates, and altitude and did not correlate with the distribution of specific species of *Chara*. Previously, we analyzed the molecular differentiation between the *C. vulgaris*–*C. contraria* complex and *C. gymnophylla* using the AFLP method in the habitats of Israel and found not only distinct species differentiation [58], but also that *C. gymnophylla* is more sensitive to arid environments, since it only occurs in northern Israel [40]. This allows us to assume that *C. vulgaris* and *C. contraria* demonstrated a tolerance for arid habitat conditions.

### 2.3. The Tentative Red List for Charophytes of South and Southeastern Kazakhstan

The thorough application of all IUCN Red List criteria is impossible due to a lack of essential data. From the available records, some species distributional, biological, and ecological traits were used to suggestion a tentative Red List for the species observed (see below).



**
*Chara aspera var. aspera*
**



**Previous records**: Lake Balkhash, including the canal from Chimpek Bay to Alakul Bay, near railway station of Akbalyk, 1968–1971 (as *C. fischeri* W.Migula, [8], Lake Balkhash drainage basin without exact localities, up to 1987 [15], shallows between the dam and the road in the vicinity of the Lake Sorbulak, 2002 [8], Lake Alakol: up to 2006 [18] and 2015–2017 [8].

**Estimation of trends in abundance and occurrence:** not possible due to an absence of details from previous records and a recent survey of formerly known sites. A drastic decline of species occurrence in Lake Balkhash drainage basin below the Kapchagay Reservoir was found during 1975–1978 because of Ili River discharge regulation into this reservoir [53].

**Other evidence for evaluation of Red List Category:** shallow water species able to withstand moderate eutrophication in brackish waters of subarid regions (R. Romanov, pers. observation).

General distribution: Holarctic.

IUCN Red List Category: VU.


2.
**
*Chara aspera var. subinermis*
**



**Previous records**: Lake Balkhash drainage basin without exact localities, up to 1987 as *C. fischeri*, [17].

**Estimation of trends in abundance and occurrence:** not possible due to an absence of details from previous records and recent survey of formerly known sites.

**Other evidence for evaluation of Red List Category:** shallow water species able to withstand moderate eutrophication in brackish waters of subarid regions (R. Romanov, pers. observation). *C. aspera* var. *subinermis* could be a common species in bays of the Lake Balkhash and neighboring water bodies. Actual scale of its distribution and threats in the regions studied cannot be estimated due to a lack of data.

General distribution: Palearctic.

IUCN Red List Category: DD (data deficient).


3.
**
*Chara canescens*
**



**Previous records**: Lake Balkhash, Alakul Bay, 1909 [11], lakes Kalgan and Abisk-Kul in Ili River delta, 1953–1964 [13], Lake Balkhash near railway station of Akbalyk, 1968–1971 [8], Lake Balkhash drainage basin without exact localities, up to 1987 [17], shallows between the dam and the road in the vicinity of the Lake Sorbulak, 2002 [8].

**Estimation of trends in abundance and occurrence:** not possible because an absence of details from previous records and recent survey of formerly known sites. *C. canescens* could be a common species in bays of the Lake Balkhash and neighboring water bodies with higher salinity. Actual scale of its distribution and threats in the regions studied cannot be estimated because of small amount of data.

Other evidence for evaluation of Red List Category: unknown.

**General distribution**: Holarctic, non-native in Australia.

IUCN Red List Category: DD.


4.
**
*Chara contraria*
**



**Previous records**: small lake in the vicinity of the village of Aksu of Almaty Oblast, 1928 [19,20], water body near road in the vicinity of the village of Kabanbay (formerly Andreevka), 1928 (var. *hispidula* A. Braun; [20,21], Lake Kalgan in Ili River delta, 1953–1964 [13], Lake Balkhash drainage basin without exact localities, up to 1987 [17], swamps in floodplain of the Charyn River in the vicinity of Sartogay relict ash grove, 2003–2005 [58], lower reaches of the Kurshilik (Kurshelek) River, 2003–2005 [59,60], lower reaches of rivers Charyn and Shilik, 2003–2005 [8], many sites in rivers Shar and Kokpekty, up to 2014 [61].

**Estimation of trends in abundance and occurrence:** not possible due to an absence of details from previous records and recent survey of formerly known sites.

**Other evidence for evaluation of Red List Category:** This is one of the most common, generalist species in many temperate regions [42].

General distribution: subcosmopolite.

IUCN Red List Category: LC.


5.
**
*Chara dominii*
**



**Previous records**: Lake Balkhash, shallows near Cape Sadyrbek, 1968–1971 [8], Kaskelen River, since 2000 [62], ponds near the settlement of Selektzii of Almaty Oblast, 2001 [8], lower reach of the Kurshilik River, 2003–2005 [59], lower reaches of rivers Charyn and Shilik, 2003–2005 [8], shallows of western part of the Lake Balkhash, 2009 [8].

**Estimation of trends in abundance and occurrence:** not possible due to an absence of details from previous records and recent survey of formerly known sites except the Kaskelen River where its populations are stable. *C. dominii* could be a common species in the bays of Lake Balkshash, Kapchagay Reservoir, and neighboring water bodies.

**Other evidence for evaluation of Red List Category:** The species had a scattered distribution in arid and semiarid regions of Eurasia [63] with most localities known from stable lakes. *C. dominii* seems to be able to form perennial stable stands in stable environments. This species is unable to grow in shallow waters (less than 0.5 m) and coarse substrates, which could explain its susceptibility to decrease water transparence as a consequence of eutrophication.

**General distribution**: Central Eurasia: Ukraine, Russia, Kazakhstan, Uzbekistan, Turkmenistan.

IUCN Red List Category: VU.


6.
**
*Chara globata*
**



**Previous records**: Lake Balkhash, water bodies of Ili River Delta, 1968–1971 (under erroneous spelling as *C. globosa*, [8,15], and Kapchagai Reservoir, 2016 [19].

**Estimation of trends in abundance and occurrence:** not possible due to an absence of details from previous records and recent survey of formerly known sites except the Kapchagay Reservoir where recent populations were found. *C. globata* could be a common species in bays of the Lake Balkhash, the Kapchagay Reservoir, and neighboring water bodies.

**Other evidence for evaluation of Red List Category:** The species has a scattered distribution in arid and semiarid regions of Eurasia and North Africa with most localities known from stable lakes [20,64]. *C. globata* seems to be able to form perennial stable stands in stable environment. Some localities in Middle East are lost [64]. This species is unable to grow in shallow waters (less than 0.5 m) and coarse substrates, which could explain its susceptibility to decrease water transparence as a consequence of eutrophication.

**General distribution**: arid and semiarid regions of Eurasia (Romania, Ukraine, Russia, Kazakhstan, Uzbekistan, Kyrgyzstan, Egypt (Sinai), Israel, Iran, China), North Africa (Tunisia, Egypt).

IUCN Red List Category: VU.


7.
**
*Chara kirghisorum*
**



**Previous records**: Lake Balkhash, shallows near Cape Sadyrbek and near railway station of Akbalyk, 1968–1971 [8], Lake Balkhash and neighboring water bodies, without exact localities, up to 1987 [17].

**Estimation of trends in abundance and occurrence:** not possible due to an absence of details from previous records and recent survey of formerly known sites.

**Other evidence for evaluation of Red List Category:** Globally, a rare species with limited distribution and few localities known.

**General distribution**: Central Eurasia: Russia, Kazakhstan, Uzbekistan, Iran, few localities in each region.

IUCN Red List Category: EN.


8.
**
*Chara tomentosa Linnaeus*
**



**Previous records**: Lake Obish-Kul in Ili River delta, 1953–1964 [13], Lake Balkhash, incl. the canal from Chimpek Bay to Alakul Bay, shallows near Cape Sadyrbek, eastern bay of Karakul Bay, Vostochny Chemyshkul Bay, 1968–1971 [8], Lake Balkhash and neighboring water bodies, without exact localities, up to 1987 [17].

**Estimation of trends in abundance and occurrence:** not possible due to an absence of details from previous records and recent survey of formerly known sites.

**Other evidence for evaluation of Red List Category:** This species seems to be able to form perennial stable stands in stable environment and in arid regions. *C. tomentosa* could be a common species in bays from Lake Balkshash, Kapchagay Reservoir, and neighboring water bodies. Actual scale of its distribution and threats in the regions cannot be estimated.

General distribution: Palearctic.

IUCN Red List Category: DD.


9.
**
*Chara vulgaris*
**



**Previous records**: Emil River in Almaty Oblast, 1842 [9], “Bunak”, 1908, and the stream of Kuchata, 1908, both in Turkestan Oblast (in former Shimkent Uezd, as *C. foetida* A.Braun [11], small lake in the vicinity of the village of Aksu (formerly Aksuyskoe) of Almaty Oblast, 1928 [20,21], Lake Balkhash and neighboring water bodies, without exact localities, up to 1987 [17], ponds near the settlement of Selektzii of Almaty Oblast, 2001 [8], shallows between the dam and the road in the vicinity of the Lake Sorbulak, 2002 [8], swamps in floodplain of the Charyn River in the vicinity of Sartogay relict ash grove, 2003–2005 [58,59], rivers Bolshaya Almatinka and Kaskelen, since 2000 [62], lower reach of the Kurshilik (Kurshelek) River, 2003–2005 [60], lower reaches of rivers Charyn and Shilik, 2003–2005 [8,18], many sites in rivers Shar and Kokpekty, up to 2014 [61], Lake Alakol, 2015–2017 [8], Kakpaktas River, 2015–2017 [65,66].

**Estimation of trends in abundance and occurrence:** not possible due to an absence of details from previous records and recent survey of formerly known sites.

**Other evidence for evaluation of Red List Category:** The most common species in Central Asia is able to grow in a wide spectrum of habitats including newly created and maintained for millennia for irrigation in the region studied [63]. This is one of the common, generalist species from many temperate regions [42].

General distribution: cosmopolite.

IUCN Red List Category: LC.

General distribution: cosmopolite.


10.
**
*Nitella hyalina*
**



**Previous records**: Ayaguz River, near mouth, 1890 [10], Lake Balkhash, without exact locality, 1953–1964 [11], Lake Balkhash and neighboring water bodies, without exact localities, up to 1987 [17], Lake Alakol: up to 2006 [18], and 2015–2017 [8].

**Estimation of trends in abundance and occurrence:** not possible due to an absence of details from previous records and recent survey of formerly known sites. Stable presence seems to be confirmed for Ili River delta and the Alakol Lake.

**Other evidence for evaluation of Red List Category:** All species of *Nitella* are really rare in Central Asia [63].

**General distribution**: cosmopolite, but really rare in many regions.

IUCN Red List Category: VU.


11.
**
*Nitellopsis obtusa*
**



**Previous records**: lakes Kara-Kultuk and Kara-Kul in low reach of Ili River, 1953–1964 [13], Lake Balkhash, incl. the canal from Chimpek Bay to Alakul Bay, Maytan Bay, 1968–1971 [8], Lake Balkhash and neighboring water bodies, without exact localities, up to 1987 [17], Lake Alakol, up to 2006 [18], ponds in the vicinity of the settlement of Mirnoe of Almaty Oblast, 2001 [8].

**Estimation of trends in abundance and occurrence:** not possible due to an absence of details from previous records and recent survey of formerly known sites. It was found from 1975–1978, but drastically declined in occurrence in the Lake Balkhash drainage basin below the Kapchagay Reservoir when the Ili River discharge regulation into this reservoir changed [53]. *N. obtusa* could be a common species in the Ili River delta lakes, neighboring bays of Lake Balkshash and the Kapchagay Reservoir.

**Other evidence for evaluation of Red List Category:** This species seems to be able to form perennial stable stands in stable environments and in arid regions. *N. obtusa* is unable to grow in shallow waters (less than 0.5 m) and coarse substrates, which could explain its susceptibility to decrease water transparence as a consequence of eutrophication [31,32].

**General distribution**: Palearctic, non-native in North America

IUCN Red List Category: VU.

## 3. Materials and Methods

### 3.1. Description of Study Site

Charophyte algae samples were collected during June-October of 2019–2022 from rivers, canals, ponds, and lakes from 3 regions (Turkestan I, Zhambyl II, and Almaty III administrative regions) (Figure 9). The surveyed localities are situated between 41.00′ to 46.40′ N and 68.12′ to 81.45′ E, at an elevation of 245–3629 m above sea level (a.s.l.) (Appendix A
Table A1).

The climate varied across the study areas [67]. Going from the southwest to the northeast direction, the average annual temperature decreased (13.2 °C in the Turkestan Region to 11.2 °C in the Zhambyl Region [68] and then to 8.6 °C in the Almaty Region) [69], while annual precipitation increased from 502.4 mm to 511.83 mm.

The regions studied include the two largest basins: the Aral-Syrdarya Basin and the Balkhash-Alakol Basin [70]. The Syrdarya and Chu rivers flow into the Aral-Syrdarya Basin. The Syrdarya River belongs in the Turkestan Region, sites 1–4 (Appendix A
Table A1). The Chu River is part of the Zhambyl Region, sites 5, 6, 8–12. Lake Mynaral, site 7, is in the south part of Zhambyl Region. The Balkhash-Alakol FEOW (Freshwater Ecoregions of the World) Basin includes the whole Ili River catchment basin. It belongs to the Almaty Region III. The studied sites are divided into three different parts of the Ili River catchment basin: 1. The first region before the Kapchagai Reservoir includes sites 42–51 and belongs to the mountainous area of the territory. 2. The second region contains sites studied from Ili River basin, including sites from the rivers Talgar (sites 35–40) and Kaskelen (sites 25–28, 30) as well as the Kapchagai Reservoir (sites 31–34). 3. The third region contains sites from the Ili River that start at site 29 after Kapchagai Reservoir dam and continue in the Lake Balkhash direction with sites in Arystan (sites 13–18), Zhidely (sites 19–22), and Bakanas (site 23) canals. There are two endorheic Lake Sorbulak (site 24) and Kurti River (site 41) belonging to the Almaty Region III. Lake Alakol (sites 52, 53, and 54) is in the eastern Almaty Region III and within a paleo-basin of Lake Balkhash, but currently there is not a connection between them.

Thus, we found that the studied habitats in the southern part of Kazakhstan are climatically like the semi-arid area of the eastern Mediterranean, and therefore our data on the environment and diversity of charophytes can be used to compare key species using methods applied for both regions.

### 3.2. Sampling and Laboratory Study

Temperature and pH were measured at the same time of sampling with a Waterproof Portable pH/Temperature meter HI991001 (HANNA instruments, USA) at Kazakhstan sites, and while in Israel conductivity and total dissolved solids (TDS) were measured with a HANNA HI 9813-0, and N-NO_3_ with a HANNA HI 93728 (HANNA Instruments, USA) with three repetitions. GPS coordinates for the sampling sites were obtained with a GARNMIN GISMAP 64. The air temperature was measured with a standard thermometer.

Dissolved oxygen, biological oxygen demand (BOD), and water color (Pt/Co scale) data were taken from the reference [8], and the monthly data from the Ministry of Ecology, Geology, and Natural Resources website of the Republic of Kazakhstan Department of Environmental Monitoring RSE “Kazhydromet”. The environmental data were defined according to [71]. The data from the documents cited in [8] were taken during the same month and year that charophytes were sampled.

Charophytes were collected in situ. Usually, charophyte mats were visible at a depth of 0–0.5 m. Each point where visible aggregations of charophytes were found was designated as a sampling point and GPS coordinates were recorded. Thus, several sampling points were established in some water bodies. Samples of charophytes were collected in a ten-meter radius from each sampling point by scrapping with anchor tugging and pulling by hands at a depth of 0–0.5 m in 5–10 samples. Samples were dried, transported to the laboratory, and labelled for permanent deposition in the Herbarium at the Institute of Botany and Phytointroduction (Kazakhstan) (label # AA 1-1 to AA 54-1). Samples were studied at the Institute of Botany and Phytointroduction, the Institute of Evolution, University of Haifa (Israel); and the Arkansas State University Beebe, Beebe in Arkansas (USA). The MBS-9 stereomicroscope (SCOPICA, Russian Federation), MicroOptix light microscope), and Leica DM2500 light microscope were used for species identification. The dimensions were taken with a microscopic eyepiece micrometer at 400–1000× magnification. The specimens were photographed with a modern Motic BA-400 microscope (Motic Asia, Hong Kong, China) and OMAX 9.0 MP USB Digital Camera. A thick layer of calcium carbonate did hamper a few of the morphological investigations by covering the plant. These specimens were treated with 4% acetic acid to dissolve the CaCO_3_.

The most relevant taxonomic reference books were used for identification of taxa [72,73,74,75]. Taxon names were checked according to the Algaebase.org website [1] for synonyms and updated.

The Ecoregions mapping program was used to create individual species distributions [56]. Statistica 12.0 was used to create maps that reflect the probability of mapped variable distribution over the lake surface according to parameter values, geospatial coordinates, and the environmental variables, which were partly measured by us and partly from reference [8] for each site [57].

Additionally, the saprobic index S was used, which describes the organic matter pollution and ecosystem state. Saprobity indices were obtained for each algal community as a function of the number of saprobic species and their relative abundances as described earlier [8]:$$S=\overset{n}{\underset{i=1}{\sum }}\left(s_{i}h_{i}\right)/\overset{n}{\underset{i=1}{\sum }}\left(h_{i}\right)$$
where *S* is index of saprobity for algal community (unitless) according to Sládeček [76], *s* is species-specific saprobity index, and *n* is the cell density of each species (Appendix A
Table A2).

The linear ordination method redundancy detrended analysis (RDA) was processed in CANOCO 4.5 program to determine the importance of main environmental factors to the species [77]. The analysis of environmental data was performed only for those 34 sites where complete environmental data were available. Species ecological preferences of studied charophytes were taken from reference [78].

The conservation status and rarity of species was assessed according to the IUCN (The International Union for Conservation of Nature) criteria [79] using the following scheme (Figure 10):

Species were assessed using five criteria [79] according to species distribution range, population size, and population change, in combination with extinction probability assessment. These criteria determined which category was most significant for each species.

#### 3.2.1. DNA Barcoding

The purpose of the polyphasic approach [54,55] was to determine the species. The complete genomic DNA was extracted from dry material taken from herbarium specimens using the standard DNeasy Plant Mini Kit (Geneaid Biotech Ltd., New Taipei, Taiwan) according to the manufacturer’s instructions. Species sequences from the Kazakhstan populations were compared to populations in Israel that are from similar climates. This specimen choice was determined by the availability of samples and the laboratory, in which the study was conducted.

Amplification of the *mat*K gene region was performed using 2× Taq Mix Red PCR MasterMix with advanced hot-start technology (PCR Biosystems Ltd., London, UK) using F-Chara (AGAATGAGCTTAAACAAGGAT) and R-Chara (ACGATTTGAACATCCACTATAATA) primers. For each PCR product, both strands were sequenced on an Applied Biosystems VeritiTM Thermal Cycler Genetic Analyzer (Applied Biosystems, CA, USA). PCR was performed with an initial two-minute denaturation step at 95 °C and one minute each for denaturation (95 °C), annealing (56 °C), and polymerization (72 °C) for 10 cycles, followed by one-minute denaturation (95 °C), annealing (52 °C), and polymerization (72 °C) for 25 cycles before the last elongation step (10 min). PCR products were visualized by 1.5% agarose gel electrophoresis with GelRed staining (GelRed^®^ Nucleic Acid Gel Stain (Biotium, Fremont, CA, USA)) and UV illumination. PCR products were purified using the Wizard ® SV Gel and PCR Clean-up Systems (Promega, Promega Corporation, Madison, WA USA) kit.

Sequencing was performed using a 3730 DNA Analyzer (Applied Biosystems, Headquarters, Thermo Fisher Scientific, Waltham, MA, USA) with identical sequencing primers to those used for PCR reactions.

#### 3.2.2. Phylogenetic Analysis

Sequences were analyzed and aligned using the BioEdit sequence alignment editor (version 7.2). The resulting sequences were corrected manually. For phylogenetic analysis, we used the *mat*K kit containing 8 *Chara vulgaris* specimens (7 specimens from Israel and 1 specimen from Kazakhstan); 4 specimens of *C. gymnophylla* (A.Braun) A.Braun from Israel and 1 specimen of *C. contraria* from Kazakhstan. The datasets were analyzed using maximum likelihood (ML), maximum parsimony (MP), and distance (neighbor connection (NJ)) in MAFFT (version 7). Using the BLAST program, areas of similarity between the obtained nucleotide sequences were found and compared with the NCBI (National Center for Biological Information) sequence databases, and statistical significance was calculated. The UPGMA (unweighted pair group method with arithmetic) clustering method was used for phylogenetic tree construction.

Data concerning morphology, *mat*K sequences, and environmental variables from a climatically similar region of Israel were used to confirm the identity *Chara contraria*, *C. vulgaris*, and *C. gymnophylla*.

## 4. Discussion

Ten charophyte species and one variety were confirmed from the regions studied. Some other species, namely *Chara aculeolata* Kütz. in Rchb. (as *C. polyacantha* A.Braun ex A.Braun, Rabenh. & Stizenb.), *C. altaica* (as *C. sibirica* W.Migula), *C. baltica* (Hartman) Bruzelius, *C. connivens* Salzm. ex A.Braun, *C. canescentiformis* Hollerbach (as *C. crinitoides* Hollerbach), *C. fragifera*, *C. galioides*, *C. globularis* Thuill. (as *C. fragilis* Desv.), *C. gymnophylla*, *C. hispida* L., *C. papillosa* Kütz. (as *C. intermedia* A. Braun ex A. Braun, Rabenh. & Stizenb.), *C. neglecta* Hollerbach, *C. schaffneri* A.Braun, *C. strigosa* A.Braun, *C. uzbekistanica* Hollerbach, *Nitella confervacea* (Bréb.) A.Braun ex Leonh., *N. tenuissima* (Desv.) Kütz., *Lychnothamnus barbatus* (Meyen) Leonh., *Lamprothamnium papulosum* (Wallr.) J. Groves were reported from this area too [7,10,13,14,15,16,17,18,19,20,23,53] but not identified during our study. Most of their records are known from the delta of the Ili River without any environmental or location details in many cases [17,53].

The charophyte abundance across the study region was not totally confirmed during our survey because previous known sites were not rechecked. At least four of them, *C. globularis*, *C. gymnophylla*, *C. papillosa* (as *C. aculeolata* sensu Hollerbach et Krassavina), and *N. tenuissima*, were found in previous studies by the authors [8,61,65]. *Lychnothamnus barbatus* was confirmed with the specimen studied [20]. In addition, *C. globata* was found in the Kapchagay Reservoir [19]. Some published species records can be misidentified [8,23] (cf. Jumakhanova et al., 2021 [23] and this work). The presence of *C. aculeolata*, *C. baltica*, *C. fragifera*, *C. galioides*, *C. hispida*, *C. schaffneri*, *C. strigosa*, and *N. confervacea* is questionable from the perspective of species distribution and ecology [7,80,81,82], and Romanov (personal communication) needs confirmation. Therefore, actual species richness before our studies could be greatly overestimated because of some misidentifications. Nevertheless, the negative trends in species richness and distribution of charophytes were noted before our studies [8,53].

A pattern was observed in that surveyed sites were inhabited by more than one species of *Chara* and only a few sites in Israel had two and more species in the same site. In our previous study, *Chara vulgaris* and *C. contraria* were shown to be distinguishable by their genome structure, whereas their morphological identification is hindered by the similarity and uncertainty of most morphological features [83]. The revealed relationship between the genome sequence and the habitat characteristics in semi-arid regions in the Israel is an example which suggests that adaptive genetic divergence in charophytes is associated with the intensity of sunlight, water level, and pH, and consequently with climatic differentiation and local environmental stresses. This plays a critical role in shaping modern charophyte diversity. Maps in Figure 3e,f demonstrate strictly the differences of both species’ distribution where *Chara vulgaris* preferred southern localities but *C. contraria* is distributed in northern sites, confirmed by RDA. As known, charophytes are usually closed to new migration, especially when an existing population becomes established in one habitat and invasive oospores compete with natives [84]. Previously, we hypothesized that site ecology is the main limiting factor for oospore establishment in a new site, meaning that gene flow is more likely to occur in sites with similar environments [83]. The discussion about the ecological preferences of these two morphologically similar species usually did not include the impact of UV radiation as a regulatory factor, but this was suggested with our observations in Israel [85]. This gives more evidence and emphasizes the importance of studying the distribution of charophytes in Kazakhstan, to have a reliable identification of both species, which is possible only with the involvement of molecular methods. In Figure 3e,f, *C. vulgaris* prefers southern sites while *C. contraria* inhabited a more northernly, less insolated sites. Therefore, the differences in distribution for these critical species are defined and confirmed by its genome sequencing and confirms the importance of species ecology. This lets us assume that the conditions for the growth of charophytes in Kazakhstan might be more favorable than for charophytes in Israel, where insolation and hydrology could regulate diversity in water bodies more strictly [83]. The cohabitation of several species of charophytes indicates a long-term non-disturbance of habitat and the fact that at least two species find it suitable, i.e., they can be excluded from the IUCN Threatened List [79]. In those cases, where only one species was found, its habitat could be severely impacted. Regarding the management of water bodies to prevent eutrophication to maintain stable charophyte stands, excessive water abstraction, salinization, and alteration of the hydrological regime as a consequence of river discharge regulation seem to be the most important threats to charophytes in the region studied. Further studies need to be conducted to evaluate the present and future stability of the ecoregions.

As a result of this study, it was possible to identify 10 species and one variety of charophyte in the studied region. As 54 sites were examined, 30 of these charophytes were found for the first time. The study of their ecology and distribution made it possible to characterize the identified species according to IUCN categories as one endangered species (*C. kirghisorum*) from a pond on the Kaskelen River, five species of the vulnerable category, and for the rest there were not enough data to determine the category.

## Figures and Tables

**Figure 2 plants-12-00368-f002:**
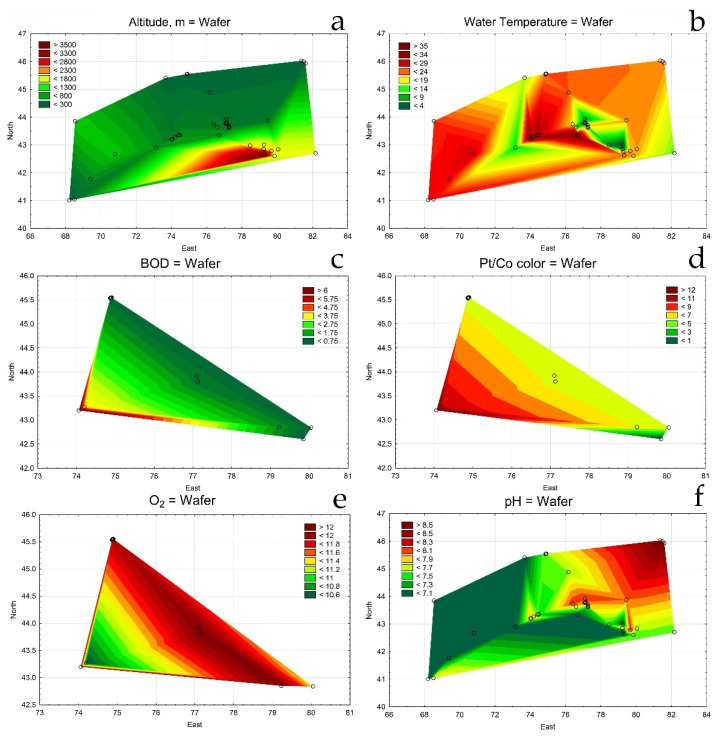
Statistical maps for environmental variables distribution on the studied area of south and southeast Kazakhstan in 2019–2022. Altitude (**a**); Water temperature (**b**); BOD (**c**); Pt/Co color (**d**); oxygen (**e**); pH (**f**). The legend key shows the variable value range.

**Figure 3 plants-12-00368-f003:**
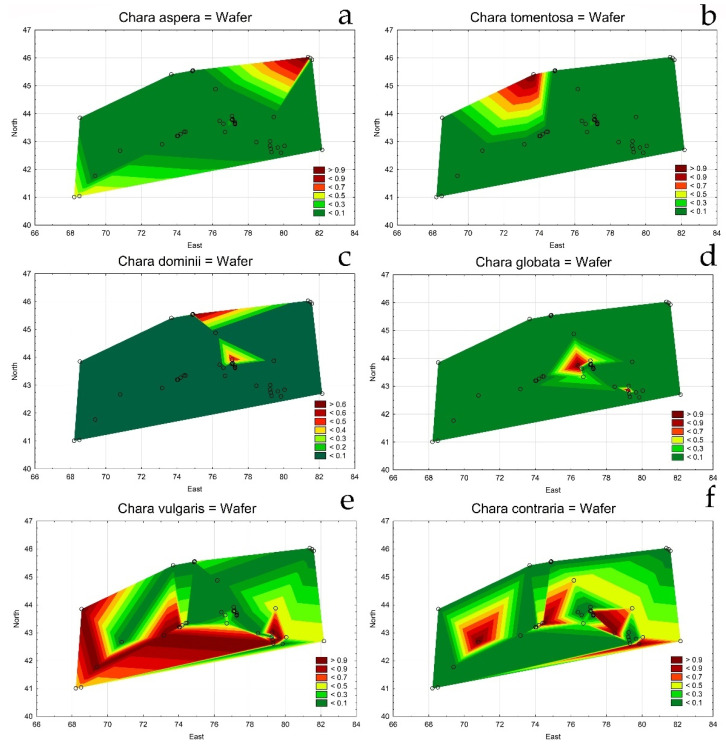
Statistical maps for *Chara* species distribution on the studied area of south and southeast Kazakhstan in 2019–2022. *Chara aspera* (**a**); *Chara tomentosa* (**b**); *Chara domini* (**c**); *Chara globata* (**d**); *Chara vulgaris* (**e**); *Chara contraria* (**f**). The legend key shows the species relative value.

**Figure 4 plants-12-00368-f004:**
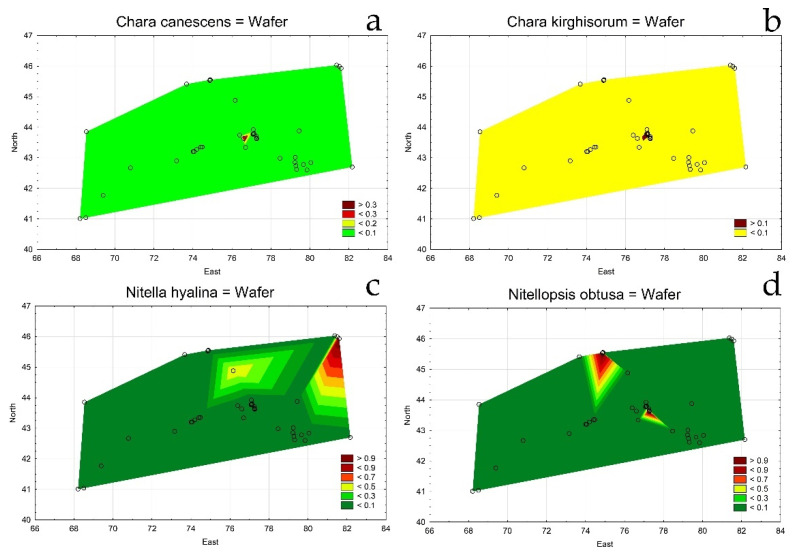
Statistical maps for species distribution on the studied area of south and southeast Kazakhstan in 2019–2022. *Chara canescens* (**a**); *Chara kirghisorum* (**b**); *Nitella hyalina* (**c**); *Nitellopsis obtusa* (**d**). The legend key shows the species relative value.

**Figure 5 plants-12-00368-f005:**
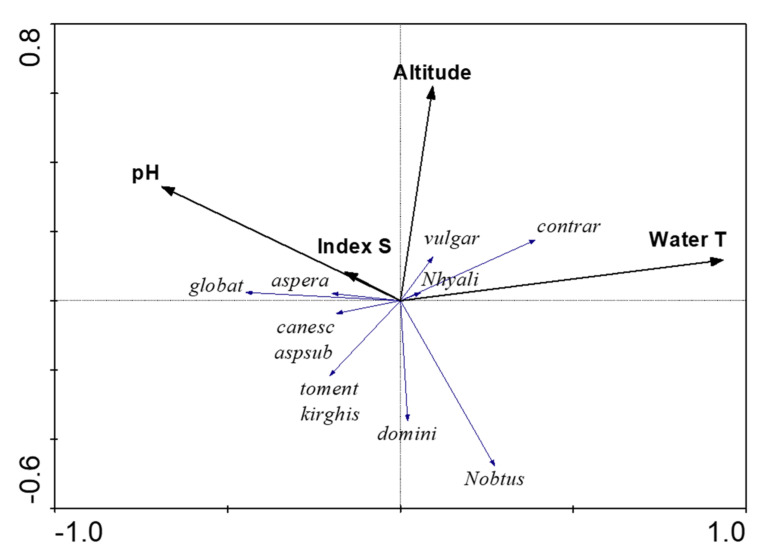
RDA plot of the relationships between environmental variables, Index of organic pollution S, altitude, pH, temperature, and species richness in the habitat community in studied sites of south and southeast Kazakhstan, 2019–2022. Monte Carlo test summary for 999 permutations: significance of first canonical axis: eigenvalue = 0.075; significance of all canonical axes: Trace = 0.177, *p*-value = 0.094.

**Figure 6 plants-12-00368-f006:**
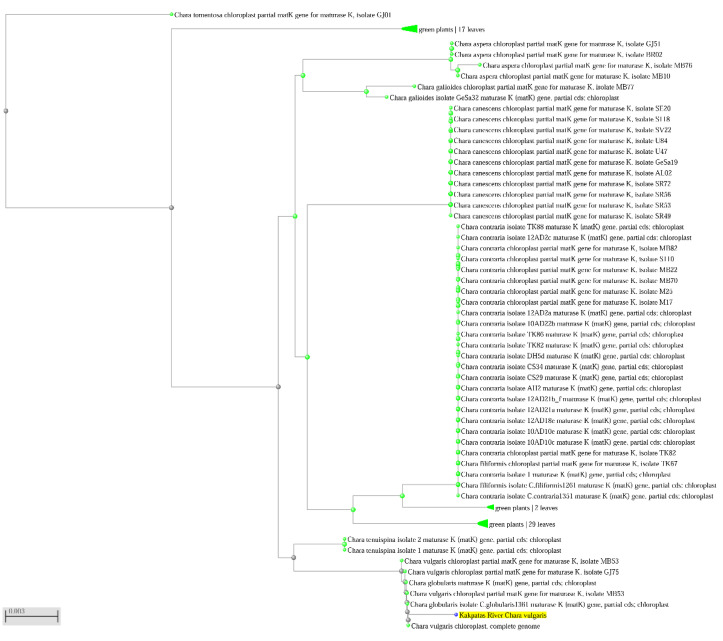
Maximum Likelihood tree of the *mat*K gene of *Chara* spp. Bootstrap values above 97 are included. The scale bar indicates 0.3% sequence divergence. Sample AA 11-2 of *Chara vulgaris* highlighted by yellow is from Kakpatas River, site 11; sequences were obtained from herbarium material. The taxon name and NSBI GenBank number are provided.

**Figure 7 plants-12-00368-f007:**
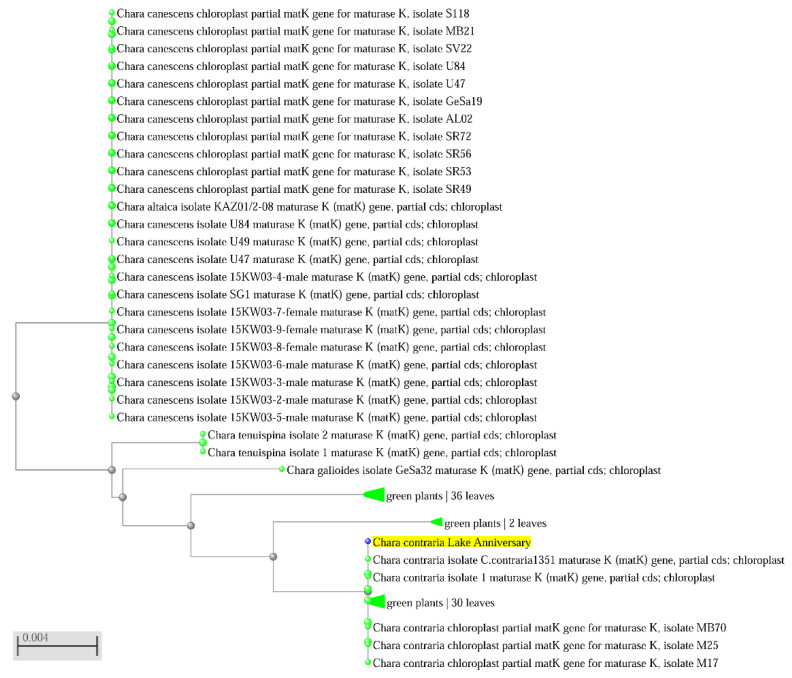
Maximum Likelihood tree of the *mat*K gene of *Chara* spp. Bootstrap values above 97 are included. The scale bar indicates 0.4% sequence divergence. Sample AA 25-1 *Chara contraria* highlighted by yellow is from The Anniversary Lake, site 25; sequences were obtained from herbarium material. The taxon name and NCBI GenBank number are provided.

**Figure 8 plants-12-00368-f008:**
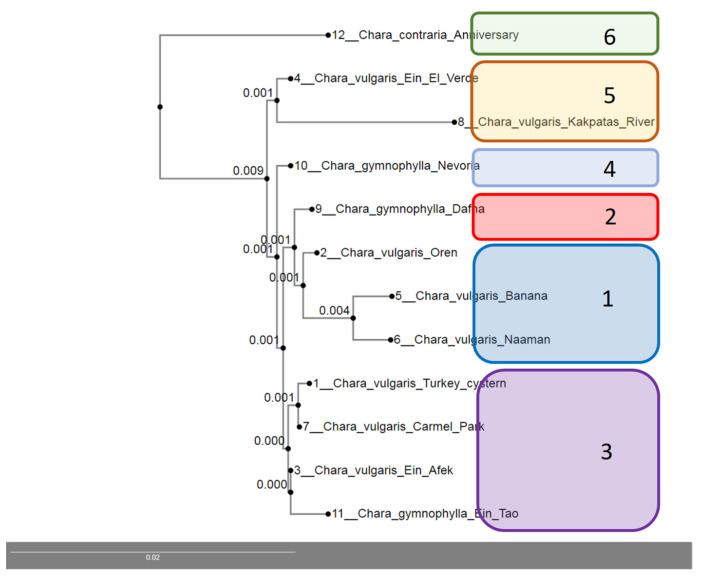
UPGMA Phylogenetic tree comparing the *mat*K sequence of charophyte species in Kazakhstan with those of Israel species. The scale bar refers to evolutionary distances in substitutions per site.

**Figure 9 plants-12-00368-f009:**
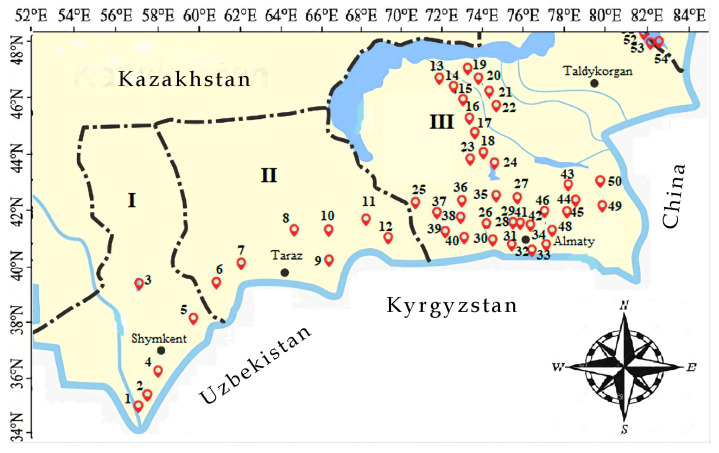
The surveyed localities (54 study sites) in southern and southeast Kazakhstan in Turkestan (I), Zhambyl (II), and Almaty (III) administrative regions in June–October of 2019–2022.

**Figure 10 plants-12-00368-f010:**
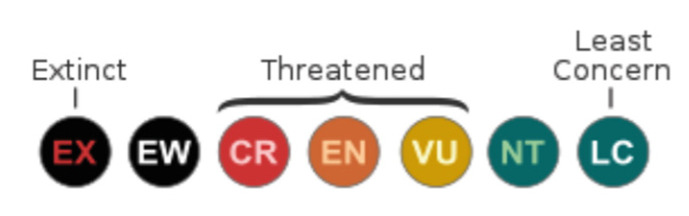
Scheme of conservation status of species according to the IUCN criteria [79].

**Table 1 plants-12-00368-t001:** Data for species environmental variables and co-occurring species of *Chara* for the studied sites of south and southeast Kazakhstan (bold) and Israel (S. Barinova own definitions).

Site Name	Species	pH	TDS, mg L^−1^	Conductivity, mSm cm^−1^	N-NO_3_, mg L^−1^	Altitude, m a.s.l.	Water T °C	Cluster	*C. contraria*	*C. globata*	*C. vulgaris*	*C. connivens* Salzmann ex. A.Braun
**Anniversery**	*C. contraria*	7.5	-	-	-	696	35.0	6	1	1	1	-
Nevoria	*C. gymnophylla*	8.0	344	0.47	2.3	690	32.2	4	-	-	-	1
Dafna	*C. gymnophylla*	7.2	240	0.33	2.5	148	23.7	2	-	-	-	-
Ein Tao	*C. gymnophylla*	7.2	421	0.58	0.9	72	24.5	3	-	-	-	-
**Kakpatas River**	*C. vulgaris*	7.0	-	-	-	561	32.0	5	1	-	1	
Ein El Verde	*C. vulgaris*	7.9	355	0.49	0.0	760	30.6	5	-	-	-	-
Oren	*C. vulgaris*	9.6	159	0.21	1.8	245	23.9	1	-	-	-	-
Banana	*C. vulgaris*	7.6	1117	1.53	1.4	0	22.3	1	-	-	-	-
Naaman	*C. vulgaris*	8.0	7815	7.78	0.7	6	31.8	1	-	-	-	-
Turkey cistern	*C. vulgaris*	7.5	318	0.44	2.2	472	20.5	3	-	-	-	-
Carmel park	*C. vulgaris*	7.5	182	0.25	0.9	339	20.8	3	-	-	-	-
Ein Afeq	*C. vulgaris*	9.4	956	1.32	1.0	13	20.0	3	-	-	-	-

## Data Availability

Not applicable.

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
