# Peer review of "Charophytes (Charophyceae, Charales) of South Kazakhstan: Diversity, Distribution, and Tentative Red List"

_plants, 2023, doi:10.3390/plants12020368_

Round 1

Reviewer 1 Report

The paper is concentrated on an important algal group and the work has been done with application of different methods, which allows achieving of the interesting results presented. 

Some technical mistakes have to be corrected and additions in the texts for clarifying the methods used are proposed: 

Lines 48-49: This group is monophyletic and comprises highly developed...

Line 50: Tree of life (Tree of Life), but not tree of

Lines 67-68: aulacanthic/tylacanthic, or aulanthous/tylacanthous, but not a combination of both in the same sentence

Line 71: pollination is a term for magnoliophytes, not for algae

Line 86: "for the evaluate the trends" - please, correct

Line 154: for first time is mentioned Israel for conductivity measurements. Please, add in the text before information about sampling sites in Israel and about the idea to compare data from both countries. Such sentence exists in the summary, but nothing is clarified in the text of the paper.

Line 189: Please, provide the authors name and reference for the saprobic index S

Lines 200-202: Please, clarify the methods used for establishment of the thretened status since below in the text no formulas are provided.

Figures 3-4 - pleae, clarify the meaning of Wafer

Author Response

Dear Editor and the Reviewer 1,

Thank you for your comments. Please find below my responses to each your comment.

With best regards,

Prof Sophia Barinova,

Responses to comments of the Reviewer 1

The paper is concentrated on an important algal group and the work has been done with application of different methods, which allows achieving of the interesting results presented. 

Some technical mistakes have to be corrected and additions in the texts for clarifying the methods used are proposed: 

Lines 48-49: This group is monophyletic and comprises highly developed...

Response: Improved “This group is monophyletic [2], and consisted of highly developed benthic macroalgae.”

Line 50: Tree of life (Tree of Life), but not tree of

Response: done

Lines 67-68: aulacanthic/tylacanthic, or aulanthous/tylacanthous, but not a combination of both in the same sentence

Response: improved to aulacanthous/tylacanthosus

Line 71: pollination is a term for magnoliophytes, not for algae

Response: Changed to “fertilization”

Line 86: "for the evaluate the trends" - please, correct

Response: Changed to “for the detection of the trends”

Line 154: for first time is mentioned Israel for conductivity measurements. Please, add in the text before information about sampling sites in Israel and about the idea to compare data from both

Response: To the MM part added:

Thus, we found that the studied habitats in the southern part of Kazakhstan are climatically like the semi-arid area of the Eastern Mediterranean, and therefore our data on the environment and diversity of charophytes can be used to compare key species using methods applied for both regions.

Data about morphology, matK sequences, and environmental variables from climatically similar region of Israel were used to confirm the identity Chara contraria, C. vulgaris, and C. gymnophylla.

Line 189: Please, provide the authors name and reference for the saprobic index S

Response: done as Sládeček

Lines 200-202: Please, clarify the methods used for establishment of the thretened status since below in the text no formulas are provided.

Response: Added: “Species were assessed with the help of five criteria [69] according to species distribution range, population size and population change, in combination with extinction probability assessment. These criteria determined which category was most suitable for each species”. The description of this method is described in the reference cited: 69. IUCN. IUCN Red List Categories and Criteria. Version 3.1. Second edition; IUCN: Gland, Switzerland and Cambridge, UK, 2012; 32 p.

Figures 3-4 - pleae, clarify the meaning of Wafer

Response: added to MM section as: “Statistica 12.0 was used to create maps that reflects the probability of mapped variable distribution over the lake surface according to parameter values, geospatial coordinates, and the environmental variables Statistica 12.0 was used to create maps that reflects the probability of mapped variable distribution over the lake surface according to parameter values, geospatial coordinates, and the environmental variables…”.

To capture of Figure 3 added: The legend key shows the variable value range.

To capture of Figures 4 and 5 added: The legend key shows the species relative value.

English was improved by native speaker.

Reviewer 2 Report

The work is well presented both as an introduction to the reading and in the detailed explanation of the methods used, the analyzes carried out and the results found. The graphic presentation of the analyzes carried out is clear and well elucidating and the bibliographic references are pertinent and exhaustive. The conclusions reached by the authors can be justified by the results. The work as a whole is well proposed and well done but lacking in really interesting content. It does not present any new methods of analysis, which are rather dated and not among the most current. e.g. DNA barcoding analysis was performed using a single plastid marker. To date, a good DNA barcoding analysis usually involves the use of at least a second, possibly nuclear marker, such as the ribosomal internal transcribed spacers. Ultimately the work as it is could also be accepted but in my opinion it is unattractive.

Author Response

Dear Editor and the Reviewer 2,

Thank you for your comments. Please find below my responses to each your comment.

With best regards,

Prof Sophia Barinova,

Responses to comments of the Reviewer 2

The work is well presented both as an introduction to the reading and in the detailed explanation of the methods used, the analyzes carried out and the results found. The graphic presentation of the analyzes carried out is clear and well elucidating and the bibliographic references are pertinent and exhaustive. The conclusions reached by the authors can be justified by the results. The work as a whole is well proposed and well done but lacking in really interesting content. It does not present any new methods of analysis, which are rather dated and not among the most current. e.g. DNA barcoding analysis was performed using a single plastid marker. To date, a good DNA barcoding analysis usually involves the use of at least a second, possibly nuclear marker, such as the ribosomal internal transcribed spacers. Ultimately the work as it is could also be accepted but in my opinion it is unattractive.

Response: This work is the first in this region and involves the expansion of both materials and methods, the results of which will be presented in subsequent papers.

English was improved by native speaker.
